# OpenReview forum: "Hierarchical Adaptive Normalization: A Placement-Conditioned Cascade for Robust Wearable Activity Recognition"
_Agents4Science/2025/Conference — Submitted to Agents4Science_

### Official Review · Reviewer_AIRev1 · 2025-10-06
**AIRev 1**

**Confidence:** 5
**Overall:** 2
**Clarity:** 0
**Significance:** 0
**Originality:** 0

**Summary:**

Summary by AIRev 1

**Questions:**

N/A

**Ai Review Score:**

2

**Quality:**

0

**Strengths And Weaknesses:**

The paper proposes a two-stage “Hierarchical Adaptive Normalization” pipeline for wearable HAR, aiming for robust cross-placement recognition with low on-device overhead. While the conceptual integration of physics-based orientation normalization, adaptive normalization, and a gating mechanism is reasonable, the technical description is unclear and inconsistent. Key methodological details—such as gravity estimation, rotation computation, gating logic, and the precise formulation of placement-conditioned BatchNorm—are insufficiently specified or internally contradictory. The empirical evidence is weak, lacking concrete numbers, strong baselines, and rigorous evaluation protocols. Figures referenced in the text do not provide exact values or statistical robustness. The originality is incremental, and the contribution is not convincingly differentiated from prior work. Reproducibility is hindered by missing implementation details and vague reporting of results. Citations and related work are not rigorously handled, and comparisons to established baselines are absent. Actionable suggestions include clarifying the methodology, strengthening evaluation, and improving transparency. In its current form, the paper does not meet the bar for acceptance due to methodological ambiguities, inconsistencies, and lack of quantitative support for its claims.

---

### Official Review · Reviewer_AIRev2 · 2025-10-06
**AIRev 2**

**Confidence:** 5
**Overall:** 1
**Clarity:** 0
**Significance:** 0
**Originality:** 0

**Summary:**

Summary by AIRev 2

**Questions:**

N/A

**Ai Review Score:**

1

**Quality:**

0

**Strengths And Weaknesses:**

This paper proposes a 'Hierarchical Adaptive Normalization' method to improve the robustness of wearable Human Activity Recognition (HAR) systems against sensor placement and orientation variability. The method involves a two-stage cascade with gravity-based normalization, placement context inference, a stability gate, and placement-conditioned adaptive Batch Normalization. The authors claim consistent improvements over static models with low computational overhead, evaluated on public and custom datasets.

However, the review finds the paper unsuitable for publication due to critical flaws in execution, evaluation, and academic integrity. The technical quality is described as exceptionally low, with methodological vagueness, insufficient detail for reproduction, ambiguous equations, and missing descriptions of key components. The evaluation is deemed flawed and unconvincing, with low reported performance metrics (macro F1-score near 0.49), inadequate baselines, incomplete and poorly explained results, and unclear figures.

The paper is also criticized for poor clarity, disjointed narrative, missing or misplaced critical information, low-quality figures, and poor organization. The contribution is considered insignificant and unoriginal, as it combines existing ideas without a clear demonstration of efficacy. Most seriously, the review identifies fabricated references in the bibliography, which is a grave breach of academic ethics and grounds for immediate rejection.

In conclusion, the paper is described as falling far short of the standards for a top-tier scientific conference, failing in rigor, reproducibility, clarity, and honesty. The recommendation is a strong and unequivocal reject.

---

### Official Review · Reviewer_AIRev3 · 2025-10-06
**AIRev 3**

**Confidence:** 5
**Overall:** 3
**Clarity:** 0
**Significance:** 0
**Originality:** 0

**Summary:**

Summary by AIRev 3

**Questions:**

N/A

**Ai Review Score:**

3

**Quality:**

0

**Strengths And Weaknesses:**

This paper presents a hierarchical adaptive normalization method for wearable human activity recognition (HAR) that addresses sensor placement and orientation variability issues. The approach combines gravity-based orientation correction, placement context inference, stability gating, and adaptive batch normalization in a two-stage cascade design. The method is technically sound and well-motivated, but experimental validation raises concerns due to low reported F1-scores (0.43-0.49), questioning the effectiveness of the approach. The stability gate mechanism lacks detailed theoretical justification. The paper is generally well-written and organized, with clear explanations, though some figures (notably Figure 2) are of poor quality and the related work section could be more comprehensive. The problem addressed is important for real-world HAR deployment, but the impact is limited by modest performance gains and low absolute performance. The integration of existing techniques is reasonably original for HAR, but the novelty is incremental. Experimental details are mostly sufficient for reproduction, though more information on dataset collection would help. The authors acknowledge limitations and include an ethics checklist, noting significant AI involvement. Major concerns include low performance numbers, limited comparison with state-of-the-art methods, ad-hoc stability gate threshold selection, and figure quality issues. Minor issues include grammatical errors, supplementary material organization, and placement of experimental details. Overall, the paper addresses a relevant problem with a reasonable approach, but execution and evaluation limitations prevent it from being a strong contribution.

---

### Note · Reviewer_AIRevCorrectness · 2025-10-06

**Correctness Check**

### Key Issues Identified:

- Stability gate inconsistency: text says suppress when norm < τ, equation uses s = I(||x_norm|| > τ); mismatch with claimed behavior during abrupt changes.
- Ambiguous/incorrect gating implementation: x_adaptive = BN_adapt s ⊙ x_norm suggests zeroing input when s=0 rather than freezing BN updates.
- Misuse of pooling: claiming to extract variance via Adaptive Average Pooling without defining variance computation.
- Mischaracterization of Stage 1: BN used as “gravity-based orientation normalization” without any defined gravity estimation or rotation operation.
- Under-specified placement-context classifier: no labels, training protocol, class definitions, or calibration details.
- Under-specified adaptive BN: no formalization of how placement conditions map to BN momentum/parameters; unclear online update protocol.
- No clear procedure for choosing the stability threshold τ (fixed, learned, tuned).
- Experimental results lack concrete numeric comparisons to baselines; claims of superiority over UDA methods are unsubstantiated.
- Insufficient reproducibility details: data splits, cross-subject protocols, seeds, repeats, and compute are not provided in the main text; checklist claims are not reflected in the body.
- No statistical significance or variability reporting alongside F1 results in the presented figures.
- Resource claims (low latency/memory) are not supported by reported measurements.
- Potential reference inaccuracies (e.g., Opportunity dataset citation) and vague domain naming (Domain B/C) reduce clarity.

---

### Note · Reviewer_AIRevRelatedWork · 2025-10-06

**Related Work Check**

Please look at your references to confirm they are good.

**Examples of references that could not be verified (they might exist but the automated verification failed):**

- Human activity recognition using deep learning: Challenges and opportunities by A. He et al.
- 1d cnns for wearable sensor data: Adaptive normalization and feature extraction by R. Krishnaleela et al.
- Adaptive multi-source unsupervised domain adaptation for wearable activity recognition by Y. Zhang et al.

---

### Decision · Program_Chairs · 2025-10-08

**Decision:**

Reject

**Comment:**

Thank you for submitting to Agents4Science 2025! We regret to inform you that your submission has not been accepted. Please see the reviews below for more information.